# Optogenetic control of gut bacterial metabolism to promote longevity

Lucas A Hartsough[1], Mooncheol Park[2], Matthew V Kotlajich[1], John Tyler Lazar[3], Bing Han[2†], Chih-Chun J Lin[2,4‡], Elena Musteata[5], Lauren Gambill[5], Meng C Wang[2,4,6]*, Jeffrey J Tabor[1,5,7]*

[1]Department of Bioengineering, Houston, United States; [2]Huffington Center on Aging, Houston, United States; [3]Department of Chemical and Biomolecular Engineering, Houston, United States; [4]Department of Molecular & Human Genetics, Baylor College of Medicine, Houston, United States; [5]Systems, Synthetic, and Physical Biology Program, Rice University, Houston, United States; [6]Howard Hughes Medical Institute, Houston, United States; [7]Department of Biosciences, Houston, United States

**\*For correspondence:**
wmeng@bcm.edu (MCW);
jeff.tabor@gmail.com (JJT)

**Present address:** [†]Children's Hospital & Institutes of Biomedical Sciences, Fudan University, Shanghai, China; [‡]Department of Molecular Biology and Genetics, Cornell University, Ithaca, United States

**Competing interests:** The authors declare that no competing interests exist.

**Abstract** Gut microbial metabolism is associated with host longevity. However, because it requires direct manipulation of microbial metabolism in situ, establishing a causal link between these two processes remains challenging. We demonstrate an optogenetic method to control gene expression and metabolite production from bacteria residing in the host gut. We genetically engineer an *Escherichia coli* strain that secretes colanic acid (CA) under the quantitative control of light. Using this optogenetically-controlled strain to induce CA production directly in the *Caenorhabditis elegans* gut, we reveal the local effect of CA in protecting intestinal mitochondria from stress-induced hyper-fragmentation. We also demonstrate that the lifespan-extending effect of this strain is positively correlated with the intensity of green light, indicating a dose-dependent CA benefit on the host. Thus, optogenetics can be used to achieve quantitative and temporal control of gut bacterial metabolism in order to reveal its local and systemic effects on host health and aging.

## Introduction

Microbiome studies have identified correlations between bacteria and host aging (*Kundu et al., 2017*; *O'Toole and Jeffery, 2015*). For example, 16S rRNA and metagenomic DNA sequencing are used to associate the presence or abundance of specific bacteria to human centenarians (*Biagi et al., 2016*; *Claesson et al., 2012*; *Claesson et al., 2011*). However, given the complexity and heterogeneity of the human gut environment, these approaches are unable to elucidate how a specific microbial species contributes to longevity.

The nematode *Caenorhabditis elegans* has a short and easily-measured lifespan, features that have revolutionized our understanding of the molecular genetics of aging and longevity (*Kenyon, 2010*). Studies using *C. elegans* also provide mechanistic insight into the association between bacterial species and host longevity (*Gusarov et al., 2013*; *Kim, 2013*). Importantly, recent studies have revealed that bacterial metabolism can produce specific products to directly influence the aging process in the host *C. elegans* or modulate the effects of environmental cues on *C. elegans* lifespan (*Cabreiro et al., 2013*; *Pryor et al., 2019*; *Virk et al., 2016*). These findings highlight the significance of bacterial metabolism in regulating host physiology during the aging process and have inspired interest in directly manipulating bacterial metabolism in situ in the host gastrointestinal (GI) tract.

In several recent studies, researchers have administered antibiotic- or carbohydrate-based small molecule inducers to modulate gene expression from gut bacteria (*Kotula et al., 2014*; *Lim et al., 2017*; *Mimee et al., 2015*). While this approach can be used to perform in situ analyses of gut bacterial-host interactions (*Lim et al., 2017*), chemical effectors may have unwanted side-effects on host or microbial physiology and are subject to slow and poorly-controlled transport and degradation processes that ultimately limit their precision.

Optogenetics combines light and genetically-engineered photoreceptors to enable unrivaled control of biological processes (*Olson and Tabor, 2014*). Previously, we and others have engineered photoreceptors that regulate bacterial gene expression in response to specific wavelengths of light (*Levskaya et al., 2005*; *Li et al., 2020*; *Ohlendorf et al., 2012*; *Ong and Tabor, 2018*; *Ong et al., 2018*; *Ramakrishnan and Tabor, 2016*; *Ryu and Gomelsky, 2014*; *Schmidl et al., 2014*). These photoreceptors have been used to achieve precise quantitative (*Olson et al., 2017*; *Olson et al., 2014*), temporal (*Chait et al., 2017*; *Milias-Argeitis et al., 2016*; *Olson et al., 2017*; *Olson et al., 2014*), and spatial (*Chait et al., 2017*; *Levskaya et al., 2005*; *Ohlendorf et al., 2012*; *Tabor et al., 2011*) control of bacterial gene expression in in vitro culture conditions. They have also been used to characterize and control transcriptional regulatory circuits (*Chait et al., 2017*; *Olson et al., 2014*; *Tabor et al., 2009*) and bacterial metabolic pathways (*Fernandez-Rodriguez et al., 2017*; *Tandar et al., 2019*) in vitro. Here, we hypothesized that optogenetic control of bacterial gene expression might provide a new way to fine-tune bacterial metabolism in the host GI tract with high temporal precision and no unwanted side-effects.

We address this possibility using the *E. coli* and *C. elegans* interaction model, an optically transparent system with known mechanistic links between bacterial metabolism and host longevity. In particular, CA is an exopolysaccharide synthesized and secreted by *E. coli* that can extend the lifespan of the host *C. elegans* by modulating mitochondrial dynamics (*Han et al., 2017*). First, we engineer *E. coli* strains that express fluorescent reporter proteins under control of our previously engineered green light-activated, red light de-activated two-component system CcaSR (*Schmidl et al., 2014*). We then expose worms carrying these engineered *E. coli* in their GI tracts to time-varying green and red light signals to demonstrate optical activation and de-activation of bacterial gene expression in worm gut. Next, we genetically engineer an *E. coli* strain that biosynthesizes CA under control of CcaSR, and utilize green light to induce CA production from this strain in the gut of the host *C. elegans.* Our experiments reveal that green light-induced CA from bacteria residing in the host gut is sufficient to modulate mitochondrial dynamics and lifespan. We use this approach to investigate the local effect of CA production on intestinal cells in a time-controlled manner and its systemic effect on organisms in a tunable way, neither of which is possible using the previous methods of administering purified CA to worms or feeding worms *E. coli* mutants that constitutively overproduce CA. This work paves the way for future studies of bacterial-host interactions with high temporal, spatial, and quantitative control without the shortcomings of chemical inducers.

## Results

To demonstrate optogenetic control over gut bacterial gene expression, we first engineered *E. coli* strain LH01, wherein CcaSR controls expression of superfolder green fluorescent protein (*sfgfp*), and *mcherry* is expressed constitutively to facilitate identification of the bacteria (*Figure 1a*, *Figure 1—figure supplement 1*, *Supplementary file 1a, 1b*). In LH01, green light-exposure switches CcaS to an active state in the presence of chromophore phycocyanobilin (PCB), wherein it phosphorylates the response regulator CcaR. Phosphorylated CcaR then activates transcription of *sfgfp* from the $P_{cpcG2-172}$ output promoter. Red light reverts active CcaS to the inactive form, de-activating *sfgfp* expression (*Figure 1a*).

We then reared two groups of *C. elegans* from the larval to the adult stage on plates of LH01 under red or green light, respectively (*Figure 1b*). Next, we washed away external bacteria, applied the paralyzing agent levamisole to prevent expulsion of gut contents, and transferred the worms to agar pads. Finally, we switched the applied light color from red to green (step ON), or green to red (step OFF), and used epi-fluorescence microscopy to image the resulting changes in sfGFP and mCherry fluorescence in the gut lumen over time (*Figure 1c*). In the step ON experiment, sfGFP fluorescence in the worm gut lumen starts low, begins to increase within 2 hr, and reaches a

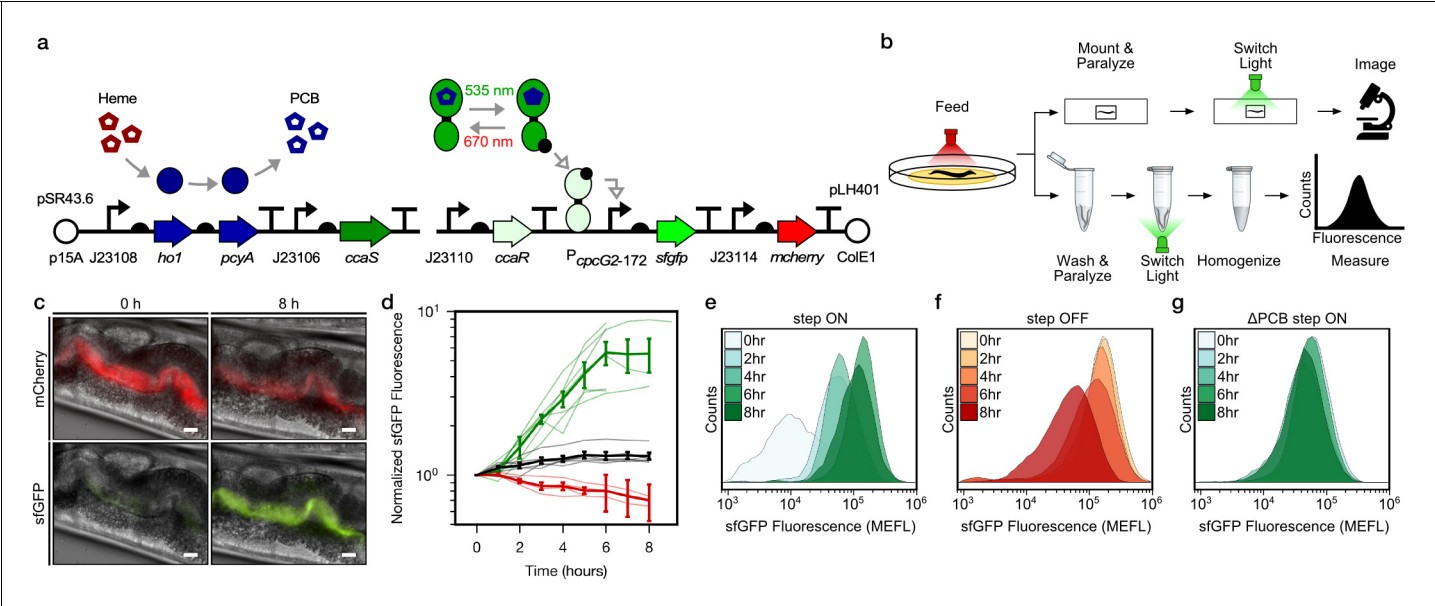

**Figure 1.** Optogenetic control of *C.elegans* gut bacterial gene expression. (**a**) Strain LH01, which harbors two plasmids: pSR43.6, which encodes the biosynthesis of the PCB chromophore, and pLH401, which encodes the CcaSR TCS with GFP as output and constitutive mCherry production. DNA element legend: origins of replication (circles), promoters (bent arrows), ribosome binding sites (semi-circles), open reading frames (block arrows), transcriptional terminators (vertical lines with flat tops). Protein legend: Dark green protein (CcaS), light green protein (CcaR), black circle (phosphoryl group). Signal legend: filled gray arrows (biochemical or biophysical process), open gray arrows (signal transduction). (**b**) Microscopy and cytometry workflows. All begin with worms fed bacteria grown in the precondition light condition. Microscopy samples were then paralyzed and imaged while exposed to the experimental light condition. Cytometry samples were washed and paralyzed, exposed to the experimental light condition, and finally homogenized before cytometry. (**c**) Fluorescence microscopy images 0 and 8 h after green light exposure in the step ON experiment. Scale bar: 10 μm. (**d**) Response dynamics in the step ON (green) and step OFF (red) microscopy experiments. Black: ΔPCB strain (step ON experiment). Individual- (light lines) and multi-worm average (dark lines) data are shown. *n* = 7, 4, 6 worms for green, red, black data sets (measured over 2, 3, 1 days, respectively). Error bars: SEM. (**e–g**) Flow cytometry histograms for response dynamics experiments. MEFL: molecules of equivalent fluorescein.

The online version of this article includes the following figure supplement(s) for figure 1:

**Figure supplement 1.** In vitro characterization of GFP reporter strains used in this study.

**Figure supplement 2.** Flow cytometry gating.

**Figure supplement 3.** Gene expression dynamics from flow cytometry experiments.

**Figure supplement 4.** Bacteria on the exterior of worms do not contribute to the measured light response in flow cytometry experiments.

saturated high level at 6 hr (*Figure 1d*). On the other hand, in the step OFF experiment, sfGFP fluorescence begins high, and decreases exponentially between hours 1–7 (*Figure 1d*). This light response is abolished when the PCB biosynthetic operon is removed (ΔPCB) (*Figure 1d*), demonstrating that changes in sfGFP fluorescence are due to CcaSR photoswitching in the worm gut.

Next, we used flow cytometry to analyze these bacterial light responses with single-cell resolution. Specifically, we reared worms on light sensing bacteria in red and green light as before, but then washed, paralyzed, and placed them into microtubes prior to light switching (*Figure 1b*). At several time points over the course of 8 hr, we homogenized the animals, harvested the gut contents, sorted bacterial cells, and measured fluorescence via flow cytometry. We observed that our engineered bacteria remain intact in the host gut (*Figure 1—figure supplement 2*) and respond to light in a unimodal fashion (*Figure 1e–g*). The temporal dynamics and PCB dependence of the gene expression responses recapitulate the microscopy results (*Figure 1—figure supplement 3*). We also confirmed that residual bacteria on the exterior of worms do not contribute to the flow cytometry measurements (*Figure 1—figure supplement 4*). Together, these microscopy and flow cytometry experiments demonstrate that we can use optogenetics to rapidly and reversibly induce gene expression of *E. coli* residing in the *C. elegans* gut.

Next, we sought to utilize our optogenetic method to modulate the production of specific metabolites in bacteria residing in the gut of live hosts. Bacterial genes involved in the same metabolic process are often clustered into operons and co-regulated at the transcriptional level. We took

advantage of this coordinated mode of regulation and chose the *cps* operon and its transcription activator RcsA for testing optogenetic control of bacterial metabolism. The *cps* operon in *E. coli* consists of 19 genes that encode enzymes required for the biosynthesis and secretion of CA (*Torres-Cabassa and Gottesman, 1987*), and CA-overproducing bacterial mutants Δ*lon* and Δ*hns* promote longevity in the host *C. elegans* (*Han et al., 2017*). To place CA biosynthesis under optogenetic control, we engineered *E. coli* strain MVK29 which lacks genomic *rcsA*, and expresses a heterologous copy of *rcsA* under the control of CcaSR (*Figure 2a*, *Supplementary file 1a, 1b*). We first examined whether MVK29 could respond to green light and induce CA production and secretion. To this end, we grew the strain in batch culture under red or green light and quantified supernatant CA levels. In red light, MVK29 secretes CA to concentrations below the limit of detection of the assay, similar to the Δ*rcsA* mutant (*Figure 2b*). Green light, on the other hand, induces MVK29 to secrete high levels of CA, and removal of the PCB biosynthetic operon abolishes this response (*Figure 2b*). Moreover, mutation of the CcaS catalytic histidine to a non-functional alanine (H534A), or the CcaR phosphorylation site from an aspartic acid to a non-functional asparagine (D51N) abolishes detectable CA production (*Figure 2b*). Importantly, the level of secreted CA increases sigmoidally with the increasing intensity of green light (*Figure 2c*), which is consistent with the response of CcaSR (*Schmidl et al., 2014*). We conclude that we have placed CA production under the control of CcaSR, and that we can use light to tune the biosynthesis and secretion of bacterial metabolites.

We then used this approach to study a gut bacterial metabolite-host interaction pathway in vivo. We first focused on cellular phenotypes in the worm gut that directly interact with bacteria and examined mitochondrial morphology that is known to be affected by CA (*Han et al., 2017*). Lon is an ATP-dependent protease that degrades RcsA, and its deletion mutant Δ*lon* shows increased CA production (*Han et al., 2017*). When grown on Δ*lon* mutant bacteria, worms exhibit elevated mitochondrial fragmentation in intestinal cells, similar to worms with CA supplementation (*Han et al.,*

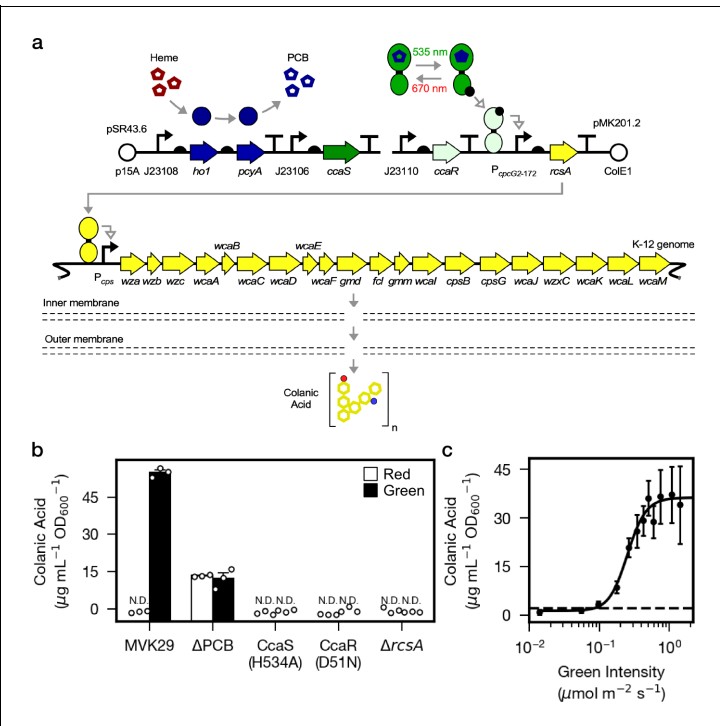

**Figure 2.** Optogenetic control of colanic acid biosynthesis. (**a**) Strain MVK29, harboring two plasmids: pSR43.6, which encodes the biosynthesis of the PCB chromophore, and pMK201.2, which includes *rcsA* under the optogenetic control of CcaSR. RcsA activates transcription of the CA biosynthetic operon, ultimately producing CA. (**b**) CA secretion levels for MVK29 and control strains exposed to red and green light. JW1935-1 is the *E. coli rcsA* background strain used in this study. N.D.: below assay limit of detection. (**c**) Green light intensity versus CA secretion level for MVK29. Data points represent three biological replicates collected on a single day. Dashed line: limit of detection. Error bars indicate standard deviation of the three biological replicates.

*2017*). To determine whether light-induced CA overproduction in MVK29 could affect the host as effectively as the CA overproduction caused by the Δ*lon* mutant, we reared two different transgenic worm strains, expressing either mitochondrially-localized GFP (mito-GFP) or mitochondrially-localized RFP (mito-RFP) in the intestine (*Supplementary file 1b*), on the plate with MVK29 and exposed the plate to red light. For each worm strain, we continued to expose one group to red light but switched a second to green for an additional 6 hr (unparalyzed light-exposure, *Figure 3a*). After paralyzing the worms with levamisole, we immediately imaged intestinal cell mitochondrial morphology using confocal microscopy and categorized the morphology into the tubular, intermediate, or fragmented (*Figure 3b*). We found that green light increases mitochondrial fragmentation in MVK29-fed worms, but not in worms fed a MVK29 ΔPCB control strain that does not respond to light (*Figure 3c*, *Figure 3—figure supplement 1*). Thus, our light-inducible CA biosynthetic strain recapitulates the effect of the Δ*lon* mutant or purified CA on the host.

We next set out to induce CA production directly from bacteria residing within the host gut and examine its effect on intestinal mitochondria. To this end, we removed worms from plates, washed away external bacteria, paralyzed them using levamisole to stop bacteria being expelled from the gut, exposed them to green or red light for 6 hr, and then imaged intestinal mitochondrial morphology using confocal microscopy (paralyzed light-exposure, *Figure 3d*). In these conditions, CA production is inactivated on the plate and worms are only exposed to CA produced from bacteria residing with the host gut. Unexpectedly, we found that 6 hr of levamisole treatment does not kill worms but leads to mitochondrial hyper-fragmentation in the intestine (*Figure 3c,e*), possibly due to the inhibitory effect of levamisole on mitochondrial NADH-oxidizing enzymes (*Köhler and*

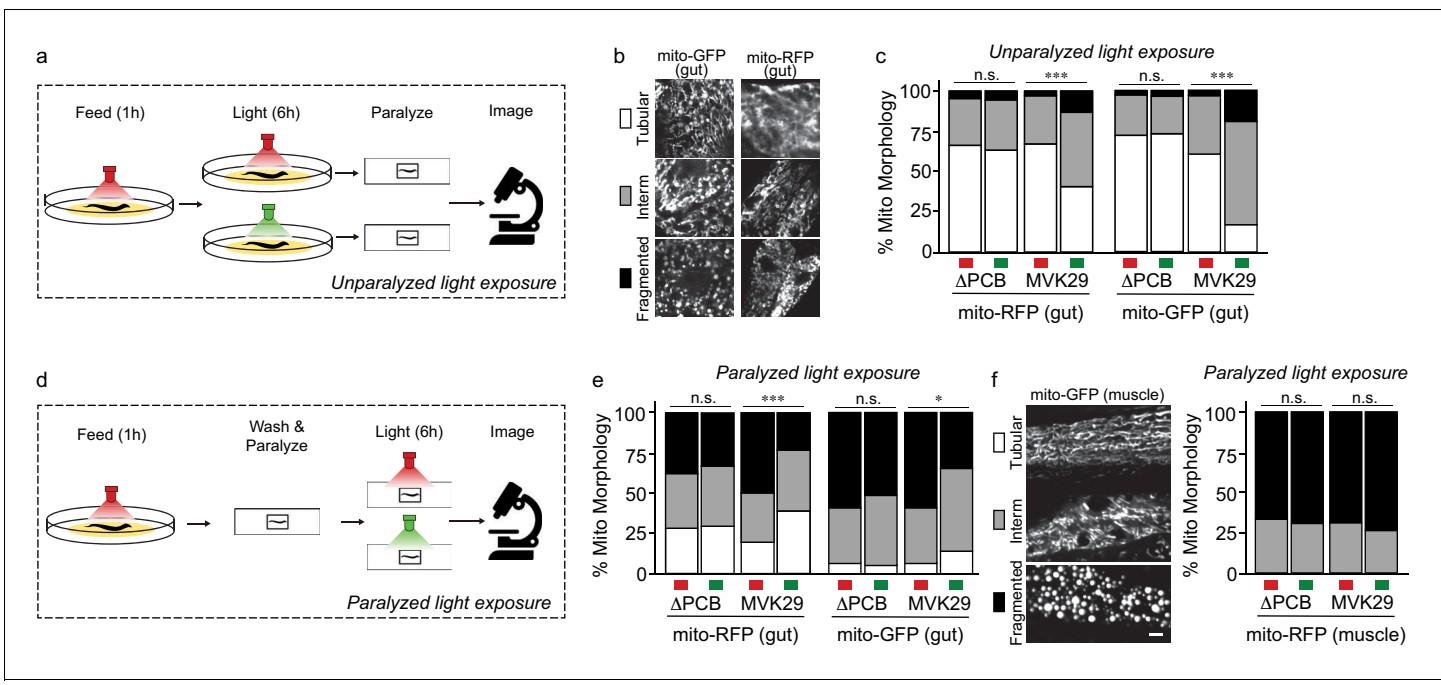

**Figure 3.** Light-regulated CA secretion modulates *C.elegans* mitochondrial dynamics. (**a**) Schematic of experiment for activating CA biosynthesis in situ using unparalyzed light-exposure conditions. (**b**) Representative images of the mitochondrial network of anterior intestinal cells visualized by either mitochondrially-localized GFP (mito-GFP) or RFP (mito-RFP) immediately distal to the pharynx are scored as fragmented, intermediate, or tubular. Scale bars: 5 μm. (**c**) Mitochondrial morphology profiles of intestinal cells in unparalyzed worms fed MVK29 while exposed to red or green light for 6 hr. (**d**) Schematic of experiment for activating CA biosynthesis in situ using paralyzed light-exposure conditions (**e**) Mitochondrial morphology profiles of intestinal cells in worms fed the indicated strain, and then paralyzed for 6 hr while exposed to red or green light. (**f**) Representative images of mitochondrial morphology in muscular cells, and the quantification of these images using worms fed the indicated strain, and then paralyzed for 6 hr while exposed to red or green light. The Chi-Squared Test of Homogeneity was used to calculate *p*-values between conditions in c, e, and f. Combined results are shown from multiple independent replicates and see *Figure 3—figure supplement 1* for data of each replicate.

The online version of this article includes the following figure supplement(s) for figure 3:

**Figure supplement 1.** Optogenetic induction of CA in situ regulates mitochondrial dynamics in the gut of *C.elegans*.

*Bachmann, 1978*) and its associated mitochondrial stress. This hyper-fragmentation phenotype was not observed in worms treated with levamisole less than 1 hr. This stress-induced fragmentation resembles mitochondrial changes related to aging and age-related neurodegenerative diseases (*Cho et al., 2009*; *Exner et al., 2007*; *Sebastián et al., 2017*). Interestingly, we found that green light exposure suppresses stress-induced hyperfragmentation in paralyzed worms bearing MVK29 in the gut (*Figure 3e*, *Figure 3—figure supplement 1*). Importantly, we found no such effects in worms bearing the ΔPCB, CcaS(H534A), CcaR(D51N) or ΔrcsA mutant strains (*Figure 3e*, *Figure 3—figure supplement 1*), suggesting that this protective effect is a result of light-induced CA overproduction in the gut. As a control using transgenic worms expressing mito-GFP in the muscle, we also examined muscular mitochondrial morphology under paralyzed light-exposure conditions. We found that 6 hr levamisole treatment induces mitochondrial hyper-fragmentation in the muscle as it does in the intestine, but that light-induced CA production from the gut bacteria does not protect muscular mitochondria against hyper-fragmentation (*Figure 3f*, *Figure 3—figure supplement 1*). These results not only show that optogenetics can be utilized to induce CA secretion from gut-borne *E. coli* in vivo, but also reveal a local protective effect of CA on intestinal cells.

Finally, we examined how the extent of light-induced CA production relates to host lifespan extension. We first confirmed that green-light-exposed MVK29 can induce CA overproduction to a level comparable to the Δlon mutant (*Figure 4a*, *Figure 4—figure supplement 1*), and neither the Δlon nor the ΔrcsA mutant responds to light (*Figure 4a*). Next, beginning at the day-1 adult stage, we exposed worms bearing MVK29 to red, or low and high green light intensities resulting in intermediate or saturating levels of CA secretion in vitro (*Figure 2c*, *Figure 4—figure supplement 1*), and measured their lifespans. We found that green light exposure extends worm lifespan compared to red light exposure (*Figure 4b*). Furthermore, the extent of lifespan extension at high green light intensity is 1.7-fold higher than that at low intensity (*Figure 4b*, *Supplementary file 1c*), suggesting a dose-dependent effect. In parallel, we measured the lifespan of worms bearing the CA-overproducing Δlon mutant and showed that the lifespan extension caused by Δlon is independent of light exposure (*Figure 4c*). The extent of lifespan extension caused by Δlon is comparable to that caused by MVK29 (*Figure 4b,c*, *Supplementary file 1c*), which is consistent with the result that Δlon and green-light-exposed MVK29 induce CA overproduction to a similar level (*Figure 4a*). In addition, the lifespan of worms bearing the ΔrcsA mutant is also light-independent and is comparable to that of MVK29 worms under red light (*Figure 4d*, *Supplementary file 1c*). These results suggest that optogenetic control is sufficient to induce bacterial production of pro-longevity compounds and improve host health. Unlike administration of a bacterial mutant, optogenetic control of bacterial metabolism can modulate a host-level phenotype in a dose-dependent manner.

## Discussion

Our method has broad applications for studying microbe-host interactions in situ. For example, we have identified about two dozen additional *E. coli* genes that are unrelated to CA biosynthesis and that enhance worm longevity when knocked out (*Han et al., 2017*), though the mechanisms by which they act remain largely unclear. By using light to induce their expression in the gut and measuring acute host responses such as changes in mitochondrial dynamics, the role of these genes in gut microbe-host interactions could be further explored. In another example, the quorum-sensing peptide CSF and nitric oxide, both of which are produced by *Bacillus subtilis* during biofilm formation, have been found to extend worm lifespan through downregulation of the insulin-like signaling pathway (*Donato et al., 2017*). We have recently ported CcaSR into *B. subtilis* and demonstrated that it enables rapid and precise control of gene expression dynamics (*Castillo-Hair et al., 2019*). The method we report here should enable optogenetic control of gene expression and metabolite production in *B. subtilis* and in situ studies of how this important Gram-positive model bacterium impacts longevity as well.

Multiple photoreceptors could also be combined to study more complex microbe-host interaction pathways. Specifically, we and others have co-expressed CcaSR with independently-controllable blue/dark and red/far-red reversible light sensors to achieve simultaneous and independent control of the expression of up to three genes in the same bacterial cell (*Fernandez-Rodriguez et al., 2017*; *Olson et al., 2017*; *Tabor et al., 2011*). Such optogenetic multiplexing could be performed in situ and used to study potential synergistic, antagonistic, or other higher-order effects of multiple

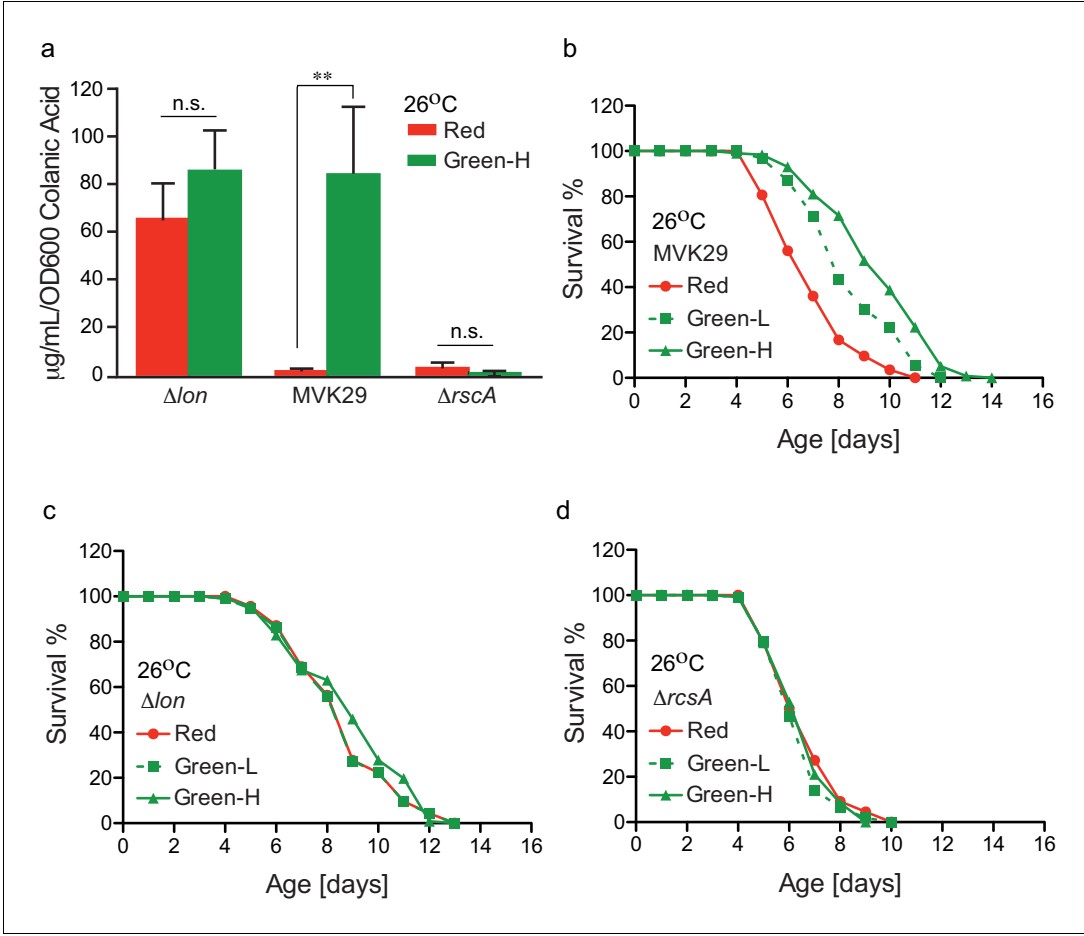

**Figure 4.** Optogenetically-regulated CA biosynthesis extends worm lifespan. (a) Green light induces CA overproduction in MVK29 to a level comparable to the Δ*lon* mutant, and neither the Δ*lon* or the Δ*rcsA* mutant changes its CA levels in response to the green light exposure. Green-H indicates a high green light intensity yielding high levels of CA production. **p<0.01 by student's *t*-test. Experiments were conducted at 26˚C. (b) When exposed to green light, worms grown on MVK29 live longer than those exposed to red light, and the magnitude of lifespan extension correlates with green light intensity (p<0.0001 green vs. red, log-rank test). Green-L indicates a low level of green light intensity resulting in intermediate levels of CA production. Experiments were conducted at 26˚C. (c–d) The lifespans of worms grown on the Δ*lon* (c) or the Δ*rcsA* (d) controls are not affected by light condition (p>0.1 green vs. red, log-rank test). Experiments were conducted at 26˚C.

The online version of this article includes the following figure supplement(s) for figure 4:

**Figure supplement 1.** Light intensity calibration.

bacterial genes or pathways. A large number of eukaryotic photoreceptors have also been developed, enabling optical control of many cell- and neurobiological processes (*Deisseroth, 2015*; *Gautier et al., 2014*; *Goglia and Toettcher, 2019*; *Leopold et al., 2018*). Bacterial and eukaryotic photoreceptors could be combined to enable simultaneous optical manipulation of bacterial and host pathways to interrogate whether or how they interact. Optogenetics could also be used to manipulate bacterial and/or host pathways at specific locations within the gut to examine location- or tissue-dependent phenomena.

Finally, optogenetic approaches could be extended to other bacteria or hosts. Although *E. coli* is an important model of Gram-negative bacteria, it is not a natural symbiont of *C. elegans*. Because *E. coli* does not stably colonize in the *C. elegans* gut, we have to paralyze worms to facilitate longer-term experiments. It would be interesting future work to port CcaSR or other bacterial photoreceptors into native *C. elegans* symbionts (*Zhang et al., 2017*) or pathogens (*Couillault and Ewbank, 2002*) and apply them for studying the physiological and pathological effects of bacterial symbionts

in vivo in non-paralyzing hosts. Moreover, it is likely that light can also be used to control gut bacterial gene expression in other model hosts such as flies, zebrafish, or mammals. Red-shifted wavelengths and corresponding optogenetic tools (*Ong et al., 2018*; *Ryu and Gomelsky, 2014*) may prove superior for less optically transparent or larger animals. Overall, by enabling precision control of bacterial gene expression and metabolism in situ, we believe that optogenetics will greatly improve our understanding of a wide range of microbe-host interactions. In particular, through aiding the digestion of ingested food, intestinal bacteria alter their gene expression and metabolism and can consequently generate specific metabolites to influence other microbes and the host in close proximity. Understanding dynamic diet-microbe-host interplays demands a quantitative method to manipulate bacterial gene expression in a temporally controlled manner. For example, upon the supplementation of a specific food source, turning on or off a specific gene in bacteria and in turn examining molecular changes in the host would help reveal primary nodes in the complex diet-microbe-host network. Given their quantitative nature, rapid and reversible switching ability, and suitability for in vivo application, optogenetic approaches will provide a new toolbox for this area of research to decipher how mechanistically microbial genetic factors participate in those diet-microbe-host interplays and contribute to host health and diseases.

# Materials and methods

## Key resources table

| Reagent type (species) or resource | Designation | Source or reference | Identifiers | Additional information |
|---|---|---|---|---|
| Recombinant DNA reagent | pLH401 | This paper | GenBank: MN617156 | Constitutive *ccaR* expression, $P_{cpcG2-172}$:*sfgfp*, constitutive *mcherry* expression |
| Recombinant DNA reagent | pLH405 | This paper | GenBank: MN617157 | $P_{cpcG2-172}$:*gfpmut3\**, constitutive *mcherry* expression |
| Recombinant DNA reagent | pLH407 | This paper | GenBank: MN617158 | Constitutive *mcherry* expression |
| Recombinant DNA reagent | pLH412 | This paper | GenBank: MN617159 | Constitutive *ccaS(H534A)*, *ho1*, and *pcyA* expression |
| Recombinant DNA reagent | pLH413 | This paper | GenBank: MN617160 | Constitutive *ccaR(D51N)* expression, $P_{cpcG2-172}$:*rcsA* |
| Recombinant DNA reagent | pSR43.6 | *Schmidl et al., 2014* | GenBank: MN617163 | Constitutive expression of *ccaS, ho1*, and *pcyA* |
| Recombinant DNA reagent | pSR49.2 | *Schmidl et al., 2014* | GenBank: MN617164 | Constitutive expression of *ccaS* |
| Recombinant DNA reagent | pMVK201.2 | This paper | GenBank: MN617161 | Constitutive expression of *ccaR*, $P_{cpcG2-172}$:*rcsA* |
| Recombinant DNA reagent | pMVK228 | This paper | GenBank: MN617162 | Constitutive expression of *ccaS* |
| Stain, strain background (*Escherichia coli*) | BW25113 | *Baba et al., 2006* | CGSC#: 7626 | Keio parent strain |
| Stain, strain background (*Escherichia coli*) | ΔrcsA | *Baba et al., 2006* | CGSC: JW1935-1 | Low CA production mutant from the Keio collection |
| Stain, strain background (*E. coli*) | Δlon | *Baba et al., 2006* | CGSC: JW0429-1 | High CA production mutant from the Keio collection |
| Genetic reagent (*C. elegans*) | zu391 | *Hermann et al., 2005* | CGC ID: JJ1271 | *glo-1*(zu391) X |
| Genetic reagent (*C. elegans*) | zcls17 | *Han et al., 2017* | CGC ID: SJ4143 | zcls17[$P_{ges-1}$::*mito-GFP*] |

*Continued on next page*

*Continued*

| Reagent type (species) or resource | Designation | Source or reference | Identifiers | Additional information |
|---|---|---|---|---|
| Genetic reagent (*C. elegans*) | *zcIs14* | *Han et al., 2017* | CGC ID: SJ4103 | *zcIs14[P$_{myo-3}$::mito-GFP]* |
| Genetic reagent (*C. elegans*) | *MW2241* | This paper | | *raxIs145[P$_{ges-1}$::mito-RFP]* |
| Genetic reagent (*C. elegans*) | *e2117* | *Han et al., 2017* | CGC ID: CB4121 | *sqt-3(e2117) V* |

## *E. coli* plasmids, strains, and media

Plasmids used in this study are described in *Supplementary file 1a*. Genbank accession numbers are given in the Key Resources Table. All plasmids constructed in this study were assembled via Golden Gate cloning (*Engler et al., 2009*). Primers were ordered from IDT (Coralville, IA). Assembled plasmids were transformed into *E. coli* NEB10β (New England Biolabs) for amplification and screening. All plasmid sequences were confirmed by Sanger sequencing (Genewiz; S. Plainfield, NJ). To construct pLH401 and pLH405, pSR58.6 (*Schmidl et al., 2014*) was modified by inserting an *mCherry* expression cassette composed of a constitutive promoter (J23114; http://parts.igem.org/Promoters/Catalog/Anderson), RBS (BBa_B0034; http://parts.igem.org/Part:BBa_B0034), *mCherry*, and a synthetic transcriptional terminator (L3S1P52 [*Chen et al., 2013*]). To construct pLH405, pLH401 was further modified by exchanging the superfolder GFP gene (*sfgfp*) for *gfpmut3\**. pMVK201.2 was built by modifying pSR58.6 to control expression of *rcsA*.

All *E. coli* strains are described in *Supplementary file 1b*. Δ*rcsA* (JW1935-1) was obtained from the Coli Genetic Stock Center. Δ*lon* (JW0429-1) was obtained from the Keio *E. coli* knockout library (*Baba et al., 2006*), a gift from the Herman lab. All *E. coli* strains were maintained in LB media supplemented with appropriate antibiotics (chloramphenicol 34 µg/mL, spectinomycin 100 µg/mL, kanamycin 100 µg/mL) in a shaking incubator at 37°C and 250 rpm unless otherwise noted.

## *C. elegans* strains and media

*C. elegans* strain information is listed in *Supplementary file 1b* and the Key Resources Table. JJ1271, SJ4143, SJ4103 and CB4121 were provided by the Caenorhabditis Genetics Center (University of Minnesota), which is funded by the NIH office of Research Infrastructure Programs (P40 OD010440). For generating the transgenic line MW2241, the N-terminal 55 amino acids sequence of *tomm-20* is fused with RFP and 3XHA tags, and the fusion is cloned into a vector containing the *ges-1* promoter. The DNA mixture is then injected into the gonad of young adult worms using the standard *C. elegans* microinjection protocol to generate transgenic strains carrying extrachromosomal arrays. The integrated transgenic strain is obtained by exposing the extrachromosomal line to 4000 rads of gamma irradiation in 5.9 min and backcrossed to wild type for more than 10 times. Worms were grown at 20°C on 1.7% NGM-agar plates in 60 mm Petri dishes inoculated with a lawn of *E. coli* (CGSC str. BW28357), as described in the CGC WormBook (wormbook.org), unless otherwise specified. The common strain *E. coli* OP50 was not used for worm feeding, as it produces approximately 50% more CA during normal growth than BW25113 (*Han et al., 2017*). M9 buffer for *C. elegans* (abbreviated M9Ce to distinguish from *E. coli* M9 media) was composed of 3 g KH$_2$PO$_4$, 6 g Na$_2$HPO$_4$, 5 g NaCl, 1 mL 1 M MgSO$_4$, H$_2$O to 1 L, and sterilized by autoclaving (wormbook.org).

## Optogenetic control of CA production

For the experiments in *Figures 2* and *3* mL starter cultures of appropriate *E. coli* strains were inoculated from a −80°C freezer and grown 12 hr at 37°C. These starters were diluted to OD$_{600}$ = 1×10$^{-2}$ in M9 minimal media (1x M9 salts, 0.4% w/v glucose, 0.2% w/v casamino acids, 2 mM MgSO$_4$, 100 µM CaCl$_2$) supplemented with appropriate antibiotics. The M9/cell mixtures were then distributed into 3 mL aliquots in 15 mL clear polystyrene culture tubes and grown at 37°C in a shaking incubator at 250 rpm while illuminated with the appropriate light wavelength and intensity, using the Light Tube Array (LTA) device (*Gerhardt et al., 2016*). After 22 hr, cultures were removed and iced to halt growth and the OD$_{600}$ was measured. Culture samples were collected for CA quantification.

For CA quantification experiments in *Figure 4* and *Figure 4—figure supplement 1*, the same preculturing, dilutions, and media conditions were utilized. However, the Light Plate Apparatus (LPA), a 24 well-plate design of the LTA (*Gerhardt et al., 2016*), was used for red and green illumination of cultures. 1 mL cultures were aliquoted into wells of a 24-well plate that corresponded with programmed red and green light intensities. Cultures grew for 22 hr in a 26°C shaking incubator at 250 rpm. Following, plates were placed on ice for 10 min, and in parallel, $OD_{600}$ for each condition was measured. Samples were then processed for CA quantification.

## CA quantification

We adapted a previous CA quantification protocol (*Dische, 1947*; *Dische and Shettles, 1948*) that takes advantage of the fact that it is the only exopolysaccharide produced in our *E. coli* strains that incorporates fucose. In particular, we quantified the amount of fucose in cell-derived exopolysaccharides (EPS), and used that value as a proxy CA levels. First, EPS was liberated from cells by boiling 2 mL of culture for 15 min. in a 15 mL conical tube. The sample was then centrifuged in 1.5 mL Eppendorf tubes for 15 min. at 21,000 x g. Then, 0.7 mL of supernatant was dialyzed against water for at least 12 hr using Pur-A-Lyzer Midi 3500 dialysis mini-tubes (Sigma-Aldrich, PURD35100-1KT) to remove monomeric fucose from the sample.

Fucose monomers were then liberated from the EPS polymers by hydrolyzing 0.2 mL of dialyzed media with 0.9 mL of $H_2SO_4$ solution (6:1 v/v acid:water). This mixture was boiled in a 15 mL conical for 20 min and then cooled to room temperature. The absorbance at 396 nm and 427 nm was measured. Next, 25 µL of 1 M L-cysteine HCl was added and mixed thoroughly by pipetting. The absorbance at 396 nm and 427 nm was measured again. Simultaneously, absorbance measurements of L-fucose standards pre- and post-L-cysteine addition were also recorded. Absorbance change, given by *D* in the formula below, were used to compare the L-fucose standard samples to the dialyzed culture samples and estimate the L-fucose concentration in the dialyzed product.

$$D = \left( \left( A^{396}_{post} - A^{396}_{pre} \right) - \left( A^{427}_{post} - A^{427}_{pre} \right) \right)$$

## Preparation of NGM-agar plates for worm feeding

3 mL *E. coli* starter cultures were inoculated from −80°C freezer stocks and grown for 12 hr at 37°C. These starters were then diluted to $OD_{600} = 1{\times}10^{-6}$ in M9 minimal media supplemented with appropriate antibiotics. The M9/cell mixture was then distributed into 3 mL aliquots in 15 mL clear polystyrene culture tubes and grown at 37°C in a shaking incubator at 250 rpm while illuminated with the appropriate light in the LTA. Once cultures reached $OD_{600} = 0.1$–0.4, tubes were iced for 10 min and subsequently concentrated to $OD_{600} \sim 20$ by centrifugation (4°C, 4000 rpm, 10 min) and resuspension in fresh M9 media. 400–600 µL of dense bacterial culture was then applied to sterile NGM-agar plates and allowed to dry in a dark room, or a room with green overhead safety lights if cultures were preconditioned in green light. Plates were wrapped in foil and refrigerated at 4°C for no more than 5 days until needed.

## Time-lapse microscopy

To obtain age-synchronized worm cultures, axenized *C. elegans* (strain *glo-1*) eggs were isolated and allowed to arrest in L1 by starvation in M9 buffer (distinct from M9 media: 3 g $KH_2PO_4$, 6 g $Na_2HPO_4$, 5 g NaCl, 1 mL 1M $MgSO_4$, and water to 1 L, sterilized by autoclaving at 121°C for 20 min) at room temperature for 12–18 hr. 10–100 larvae were transferred to a previously prepared NGM-agar plate containing a lawn of the appropriate bacterial strain. The plate was then placed in a 20°C incubator and illuminated with appropriate optogenetic light provided by a single LED positioned 1 cm above the Petri dish. Adult worms were transferred to a fresh plate as necessary to maintain only a single generation.

Individual worms aged 1–3 days were removed from the dish and prepared for time-lapse epifluorescence imaging. A 1.5% agar pad was prepared using M9 buffer as previously described (*Ryu and Gomelsky, 2014*), and punched into ½" circles with a hollow punch. A 4 µL droplet of 2 mM levamisole was deposited on a pad and five adult worms were transferred from the NGM plate to the droplet. An additional 4 µL of levamisole solution was added, and the worms were gently washed to remove external bacteria. Worms were then transferred to a fresh pad with a 4 µL droplet of levamisole solution, which was allowed to dry, thereby co-localizing and aligning the

worms on the pad. The pad was then inverted and placed into a 13 mm disposable microscopy petri dish with a #1.5 coverslip on the bottom (Cell E and G; Houston, TX). Another coverslip was placed on the top of the pad in the dish to curtail evaporation.

The dish was then placed on the stage of a Nikon Eclipse Ti-E inverted epifluorescence microscope (Nikon Instruments, Inc; Melville, NY). Complete paralysis was induced by incubating the dish at room temperature (~23°C) for 30 min. Meanwhile, worms were exposed to appropriate preconditioning light supplied by a circular array of 8 LEDs (4 × 660 nm, 4 × 525 nm) mounted to the microscope condenser ring, about 2 cm above the Petri dish. Light was then switched from the preconditioning to the experimental wavelength, and worms were imaged periodically using 10x, 40x, and 60x objectives. For each time point, the LEDs were turned off and images acquired in the brightfield (DIC) and fluorescent channels. Afterwards, the LEDs were turned on again to maintain optogenetic control.

## Epifluorescence image analysis

All epifluorescence images were analyzed using the Nikon Elements software package (Nikon Instruments, Inc; Melville, NY). The mCherry signal was used as a marker for the gut lumen, and only cells in this region were included in the analysis. Image ROIs were created by thresholding the sfGFP signal to identify the boundaries of cell populations. Out of focus regions were eliminated from analysis. The average sfGFP pixel intensity inside the ROIs was calculated and recorded for each time point.

## Flow cytometry

1–3-day-old *glo-1* worms were prepared for flow cytometry of the microbiome constituents by washing, using a protocol adapted from previous work (*Portal-Celhay et al., 2012*). Groups of 5 worms were washed 2x in a 5 µL droplet of lytic solution: *C. elegans* M9 buffer containing 2 mM levamisole, 1% Triton X-100, and 100 mg/mL ampicillin. The worms were then washed 2x in 5 µL droplets of M9 buffer containing 2 mM levamisole only. Finally, the worms were transferred to clear 0.5 mL Eppendorf tubes containing 50 µL of M9 buffer + 2 mM levamisole, ensuring that five worms were deposited in the liquid contained in each tube. Each tube was then exposed to light by placing it within one well of a 24-well plate (AWLS-303008, ArcticWhite LLC) atop a Light Plate Apparatus (LPA) containing green and red LEDs[47] for 8 hr at room temperature. In separate control experiments, we demonstrated that any stray bacteria that may escape the worms over this period, or which were inadvertently added to the 50 µL of M9 buffer, are incapable of responding to optogenetic light (*Figure 1—figure supplement 4*).

At the conclusion of the experiment, tubes were removed from the plate and immediately chilled in an ice slurry for 10 min in the dark. Worms were homogenized using an anodized steel probe sterilized between samples via 70% ethanol treatment and flame before being cooled.

Next, we used our previous antibiotic-based fluorescent protein maturation protocol (*Olson et al., 2014*) to allow unfolded proteins to mature while preventing the production of new protein. In particular, 250 µL PBS containing 500 mg/mL Rifampicin was added to the 50 µL homogenized worm samples and transferred to cytometry tubes. These tubes were incubated in a 37°C water bath for precisely 1 hr, then transferred back to an ice slurry.

These samples were measured on a BD FACScan flow cytometer. For gating, an FSC/SSC polygon gate was first created using non-fluorescent bacteria grown in vitro at 37°C (*Figure 1—figure supplement 2*). Events outside this region were excluded as non-bacterial material. To isolate the engineered gut bacteria, only events with a high mCherry signal (FL3 >1200 a.u., FL3 gain: 999) were included (*Figure 1—figure supplement 4*). Samples were measured until 20,000 events were recorded or the sample was exhausted.

## Flow cytometry data analysis

All flow cytometry data (FCS format) were analyzed using FlowCal (*Castillo-Hair et al., 2016*) and Python 2.7. We wrote a standard cytometry analysis workflow that truncated the initial and final 10 events to prevent cross-sample contamination, removed events from saturated detector bins at the ends of the detection range, and added 2D density gate on SSC/FSC retaining the densest 75% of events (*Figure 1—figure supplement 2*). GFPmut3* fluorescence units were converted into

standardized units of molecules of equivalent fluorescein (MEFL) using a fluorescent bead standard (Rainbow calibration standard; cat. no. RCP-30-20A, Spherotech, Inc) as described previously (*Castillo-Hair et al., 2016*). Finally, to eliminate events associated with *C. elegans* auto-fluorescence (*Figure 1—figure supplement 2*), any events in the region FL1 $\leq$ 1200 MEFL were discarded.

## Mitochondrial fragmentation assays

Synchronized L1 worms (strain *ges-1*) were applied to NGM-agar plates containing bacterial strain BW25113 and allowed to develop until adulthood. This parental bacterial strain is used to allow all worms to develop at the same rate, avoiding any developmental/growth effects the experimental strains may exert on the worms. All experimental bacterial strains were preconditioned in red light with optogenetic light provided by a single LED positioned 1 cm above the Petri dish.

After 3–5 days (between days 1–3 of adulthood), worms allocated for the experiment were transferred to experimental strains for approximately 60–90 min and exposed to red light to thoroughly inoculate the GI tract without inducing CcaSR. In the case of the unparalyzed worms, red or green light was then applied for an additional 6 hr. For the paralyzed worm experiments, 1.5% low-melt agar pads were prepared as described above and placed on individual slides. About 15 adult worms were transferred from the experimental strain Petri dish to an agar pad containing 10 μL of *C. elegans* M9 buffer + 2 mM levamisole (M9Ce+Lev), where worms were gently washed before being transferred to a fresh pad also containing 10 μL of M9Ce+Lev. The majority of M9Ce+Lev on the pad was allowed to evaporate, which causes the worms to align longitudinally before a cover slip was applied. Slides were then exposed to either red or green light by placing them under a single LED positioned 1 cm above the Petri dish for 6 hr. Afterward, intestinal and muscular cells were imaged using confocal microscopy (Olympus Fluoview 3000) in the brightfield and GFP/RFP channels.

## Confocal microscopy image analysis

All confocal images for an experiment were randomized and the mitochondrial networks of each were blindly classified by another researcher independently as either tubular, fragmented, or intermediate. Tubular samples were marked by a high degree of network connectivity throughout. Fragmented samples were composed almost exclusively of isolated clusters of fluorescence with high circularity. Intermediate samples contained regions of both types. Scores were then de-randomized and aggregated. For each experimental strain, the red and green light conditions were compared for statistical significance using the chi-squared test of homogeneity.

## Lifespan experiments

3 mL starter cultures of Δ*lon*, Δ*rcsA* or MVK29 were inoculated from −80°C freezer stocks into LB supplemented with appropriate antibiotics and grown shaking for 12 hr at 37°C at 250 rpm. These cultures were diluted to $OD_{600} = 1 \times 10^{-6}$ in 27 mL M9 media supplemented with appropriate antibiotics. 1.5 mL of each M9/cell mixture was added to each of 18 wells on three 24-well plates and grown in 3 LPA devices under the appropriate light conditions at 37°C and 250 rpm. Once cultures reached $OD_{600} = 0.1$–$0.4$, all tubes were iced for 10 min and subsequently concentrated 10 times by centrifugation (4°C, 4000 rpm, 10 min). Approximately, 50 μL of this dense bacterial culture was then applied to sterile NGM-agar plates with no antibiotics and allowed to dry in a dark room. The plates were then illuminated with the appropriate light wavelength and intensity for 16 hr at room temperature, and immediately used for lifespan assays.

During the longitudinal lifespan assay, exposure to white light is limited to the minimal level. To reach this goal, the *sqt-3(e2117)* temperature sensitive mutant (*Supplementary file 1b*) was used to perform longitudinal analyses at 25°C, which avoids time-consuming animal transfer without interrupting normal reproduction. *sqt-3(e2117)* is a collagen mutant of *C. elegans* that reproduces normally but is embryonically lethal at 26°C and has been used previously in longitudinal studies (*Han et al., 2017*; *Wang et al., 2014*). Worms were age-synchronized by bleach-based egg isolation followed by starvation in M9 buffer at the L1 stage for 36 hr. Synchronized L1 worms were grown on BW25113 *E. coli* at 15°C until the L4 stage, when worms were transferred to 24-well plates (~15 worms/well) with Δ*lon*, Δ*rcsA* or MVK29 (*Supplementary file 1b*). The plates were placed in LPA

devices. LPA LEDs were programmed to illuminate wells with constant red (10 μmol/m$^2$/s), low-intensity green (0.25 μmol/m$^2$/s), or high-intensity green light (10 μmol/m$^2$/s). The apparatus was then transferred to a 26˚C incubator. The number of living worms remaining in each well was counted every day. Death was indicated by total cessation of movement in response to gentle mechanical stimulation. Statistical analyses were performed with SPSS (IBM Software) using Kaplan-Meier survival analysis and the log-rank test (*Supplementary file 1c*).

## Acknowledgements

The LED array used to illuminate *C. elegans* plates was designed by Brian Landry and Sebastián Castillo-Hair. The mounting hardware for the microscope LED array was designed by Ravi Sheth. We thank Ravi Sheth for discussions during early stages of the project. We thank Dr. Joel Moake for the use of his cytometer. This work was supported by the John S Dunn Foundation (JJT and MCW), the Welch Foundation (JJT and MCW), the US National Institutes of Health 1R21NS099870-01 (JJT), DP1DK113644 (MCW), R01AT009050 (MCW), R01AG062257 (MCW), and the Howard Hughes Medical Institute (MCW). LAH was supported by a NASA Office of the Chief Technologist Space Technology Research Fellowship (NSTRF NNX11AN39H). JTL was supported by a US National Defense Science and Engineering Graduate Fellowship. EM was supported by a US National Science Foundation Graduate Research Fellowship.

## Additional information

### Funding

| Funder | Grant reference number | Author |
| --- | --- | --- |
| National Institutes of Health | 1R21NS099870-01 | Jeffrey J Tabor |
| National Institutes of Health | DP1DK113644 | Meng C Wang |
| National Institutes of Health | R01AT009050 | Meng C Wang |
| National Aeronautics and Space Administration | NSTRF NNX11AN39H | Lucas A Hartsough |
| John S. Dunn Foundation | | Meng C Wang<br>Jeffrey J Tabor |
| Welch Foundation | | Meng C Wang<br>Jeffrey J Tabor |
| National Institutes of Health | R01AG062257 | Meng C Wang |
| Howard Hughes Medical Institute | | Meng C Wang |
| National Defense Science and Engineering Graduate | | John Tyler Lazar |
| National Science Foundation | Research Fellowship | Elena Musteata |

The funders had no role in study design, data collection and interpretation, or the decision to submit the work for publication.

### Author contributions

Lucas A Hartsough, Conceptualization, Data curation, Formal analysis, Funding acquisition, Investigation, Visualization, Methodology, Writing - original draft, Writing - review and editing; Mooncheol Park, Data curation, Formal analysis, Validation, Investigation, Visualization, Methodology, Writing - review and editing; Matthew V Kotlajich, Conceptualization, Data curation, Methodology; John Tyler Lazar, Data curation, Formal analysis, Validation, Investigation, Methodology; Bing Han, Conceptualization, Validation, Investigation, Visualization, Methodology; Chih-Chun J Lin, Data curation, Formal analysis, Investigation, Visualization, Methodology; Elena Musteata, Validation; Lauren Gambill, Investigation; Meng C Wang, Conceptualization, Data curation, Formal analysis, Supervision, Funding acquisition, Writing - original draft, Writing - review and editing; Jeffrey J Tabor,

 

Conceptualization, Supervision, Funding acquisition, Investigation, Writing - original draft, Writing - review and editing

### Author ORCIDs
Lauren Gambill  http://orcid.org/0000-0002-1490-2449
Meng C Wang  https://orcid.org/0000-0002-5898-6007

### Decision letter and Author response
Decision letter https://doi.org/10.7554/eLife.56849.sa1
Author response https://doi.org/10.7554/eLife.56849.sa2

---

## Additional files

### Supplementary files
• Supplementary file 1. Supplementary Tables. (**a**) Table of plasmids used in this study. (**b**) Table of bacterial and worm strains used in this study (**c**) Statistical analysis of worm lifespan experiments

• Transparent reporting form

### Data availability
All data generated or analysed during this study are included in the manuscript and supporting files.

---

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
