## [Decision Letter]

**Acceptance summary:**

The reviewers and editors feel that the approach and techniques presented are exciting, with important future applications – especially given progress on extension the approach to e.g. *B. subtilis*. You convincingly demonstrate the utility and value of your novel optogenetic approach for investigating bacterial metabolism.

**Decision letter after peer review:**

Thank you for submitting your article "Optogenetic control of gut bacterial metabolism to promote longevity" for consideration by *eLife*. Your article has been reviewed by three peer reviewers, one of whom is a member of our Board of Reviewing Editors, and the evaluation has been overseen by Jessica Tyler as the Senior Editor. The following individual involved in review of your submission has agreed to reveal their identity: William Mair (Reviewer #2).

The reviewers have discussed the reviews with one another and the Reviewing Editor has drafted this decision to help you prepare a revised submission. We all feel that you when you resubmit the revised paper, it should be as a "Tools and Resources" paper rather than an article, given the methodological nature of your study.

Summary:

Hartsough et al. present a method to optogenetically control the production of colanic acid (CA) from a genetically engineered *E. coli* strain. Using this *E. coli* strain as a feed for *C. elegans*, the authors show CA production directly in the animal gut and its local effects in altering mitochondrial dynamics in the gut. The authors corroborate findings from a previous publication from Han et al., 2017, showing effects on *C. elegans* lifespan and intestinal mitochondrial network.

The concept of modifying host biology via optogenetic control of gut flora is novel and would be fitting for the scope of *eLife*. However, the reviewers had some concerns regarding some of the claims regarding biological insights as presented. Given the current COVID-19 situation, we feel that asking for additional experiments and novel biological findings would be beyond the scope of a revision, but the reviewers suggested that additional data or edits may help clarify how this work is distinct from Han et al., 2017, why an optogenetic approach is better than the standard genetic approaches and what the strength and weaknesses of this approach are.

Essential revisions:

1) One of the key claims made by the authors is that the optogenetic approach allows for more fine scale control of metabolite production and further that they "investigate how the lifespan extending effect of CA relates to CA levels". However, aside from one ex vivo assay (Results, Figure 2B/C), the amount of CA being secreted (or taken up by the host) is not reported. The authors claim, but don't actually show a relationship of lifespan with CA level, just green-light intensity. The relative CA production levels of the engineered strains under green light vs. the over-producing mutants should ideally be quantified and both compared to realistic exposure levels in vivo of conventionally grown *C. elegans*. If this cannot be done without additional experiments, the authors should be careful to temper their claims here.

2) A related concern is how do the authors know that the effects are not mediated by CA produced and secreted on plates. They attempt to answer this question using levamisole paralyzed worms – but the mitochondrial morphology of paralyzed worms seems very different (hyper fragmented, Figure 3D) form non-paralyzed – as the authors themselves concede. I agree that there appears to be some rescue effect in MVK29 exposed to green light – but it would be preferable to show benefits of in situ expression without the overlaying levamisole detriment.

3) The mitochondrial fragmentation results are obviously motivated by the findings in Han et al., 2017, but are not well introduced/explained. In the fifth paragraph of the Results it is stated that exposure to light-induced CA secretion increases fragmentation. But in the next paragraph that light-induced CA secretion decreases mitochondrial fragmentation caused by levamisole treatment. The interpretation of these conflicting results-and their relevance to other systems lacking stress induced mitochondrial fragmentation-is unclear and should be more clearly discussed.

4) In addition to local effects in the gut, were the authors able to see effects in distal tissues? An experiment investigating effects of light-induced CA production in the gut, affecting body wall muscle mitochondrial network similar to Han et al., 2017, could strengthen the current manuscript and show expected systemic effects of CA and not just local.

5) This work relies on a non-standard gut microbe for *C. elegans* and requires that the animals be paralyzed to maintain colonization through measurement. This is addressed only in passing in the Discussion, but altogether still gives pause. Yes, *E. coli* is "this important Gram-positive model bacterium," but it isn't in fact a very important bacterium to this host. Limitations and open questions that should be made explicitly are: Whether host uptake of metabolites produced by a symbiont would differ from that of a food source, whether/how this approach could be extended to bacteria other than *E. coli* and how such interactions might play out in a community context and whether / how optogenetics could be used to study this question.

6) As a standard *C. elegans* feeding strain, OP50 is naturally *lon* deficient. Can the authors comment on the levels of CA production from their light-induced methods as compared to the regular OP50? If no such data is available, this should still be discussed.

7) The authors make a claim regarding the relative magnitude of lifespan effects of purified CA vs. CA produced using their optogenetic approach. However, it appears that claim is based on comparison of lifespan effects with previously published results. Such comparisons are risky because differences in lab, feeding conditions, and strain. It would also have been preferable to have a CA "fed" control and it would further have been interesting to carry out a study of optogenetic induction in the presence of exogenous CA. In the absence of a CA fed control, the authors should be careful with their claims regarding effect size.

8) There were some concerns over the exclusive use of GFP with a mito targeting sequence as the reporter because this only labels mitochondria if they are functionally importing the GFP. In addition there seems to be a lot of noise from intestinal gut granules and aggregates in Figure 3B. These data could be improved if an additional label of mitochondria was used – either a membrane marker or dye, in order to ensure the changes seen are reflecting changes to mitochondrial network morphology and not GFP aggregation, or non-mitochondrial localized GFP signal. The authors should be careful to make these limitations clear.

9) Lifespan of control and treated animals in general appear shorter than would be expected. E.g. in Figure 4, the lifespan of worms fed the *E. coli* background strain JW1935-1 is considerably short. Can the authors comment on any previous studies with *C. elegans* lifespan on this strain and if this is expected? Are there other optogenetic background strains available?

---

## [Author Response]

Essential revisions:1) One of the key claims made by the authors is that the optogenetic approach allows for more fine scale control of metabolite production and further that they "investigate how the lifespan extending effect of CA relates to CA levels". However, aside from one ex vivo assay (Results, Figure 2B/C), the amount of CA being secreted (or taken up by the host) is not reported. The authors claim, but don't actually show a relationship of lifespan with CA level, just green-light intensity. The relative CA production levels of the engineered strains under green light vs. the over-producing mutants should ideally be quantified and both compared to realistic exposure levels in vivo of conventionally grown *C. elegans*. If this cannot be done without additional experiments, the authors should be careful to temper their claims here.

We appreciate the reviewer’s comment on the comparison of CA levels among different conditions. Technically, it is currently impossible to measure CA levels in bacteria grow on solid agar plates or in the worm gut, or in the host cells. Thus, our conclusion is based on the ex vivo assay using liquid culture and the positive relationship between green-light intensity and the amount of CA being secreted. Based on the reviewer’s suggestion, we have revised the text to explicitly state this relationship. In addition, we have included new results presenting a direct comparison of CA levels between the Δ*lon* mutant and the light inducible strain (new Figure 4A).

2) A related concern is how do the authors know that the effects are not mediated by CA produced and secreted on plates. They attempt to answer this question using levamisole paralyzed worms – but the mitochondrial morphology of paralyzed worms seems very different (hyper fragmented, Figure 3D) form non-paralyzed – as the authors themselves concede. I agree that there appears to be some rescue effect in MVK29 exposed to green light – but it would be preferable to show benefits of in situ expression without the overlaying levamisole detriment.

We understand the reviewer’s concern on the possible confounding effect of levamisole. We agree that levamisole is an imperfect means by which to paralyze the worms, resulting in undesired changes to host mitochondrial morphology. Unfortunately, host paralysis and cessation of host peristaltic activity is required to measure the effect of CA production in vivo; otherwise, bacteria would be expelled before sufficient amount of CA accumulated to affect mitochondrial morphology in the host. We have tried several alternative means of immobilizing the host for this experiment, and only levamisole is sufficient to stop bacteria expulsion from the gut and keeps worms alive after 6 hours of the treatment.

Regarding the concern that CA may be produced on the plate instead of in the host, in the context of Figure 3 (new Figure 3E and F), note that neither bacteria nor worms on the plate are exposed to green light (which induces CA production). Worms are only exposed to green light after they have been removed from the plate and paralyzed. Thus, we conclude that the effect on mitochondrial morphology is mediated by CA produced from bacteria inside the host gut. In the context of Figure 4, we expect that CA produced from bacteria on the plate and in the host gut both contribute to the lifespan-extending effect. Currently, we cannot experimentally separate these two processes. In the revised manuscript, we have included new diagram figure panels to illustrate the experimental design (new Figure 3A and D) and improved the language to introduce the experimental design more clearly and increase the textual clarity of the related claims.

3) The mitochondrial fragmentation results are obviously motivated by the findings in Han et al., 2017, but are not well introduced/explained. In the fifth paragraph of the Results it is stated that exposure to light-induced CA secretion increases fragmentation. But in the next paragraph that light-induced CA secretion decreases mitochondrial fragmentation caused by levamisole treatment. The interpretation of these conflicting results-and their relevance to other systems lacking stress induced mitochondrial fragmentation-is unclear and should be more clearly discussed.

Thank the reviewer for pointing out this concern. We have revised the manuscript to (1) better introduce the previously published findings, (2) explain the new results more clearly, and (3) discuss the potential difference between these two conditions.

4) In addition to local effects in the gut, were the authors able to see effects in distal tissues? An experiment investigating effects of light-induced CA production in the gut, affecting body wall muscle mitochondrial network similar to Han et al., 2017, could strengthen the current manuscript and show expected systemic effects of CA and not just local.

As the reviewer suggested, we have examined the effect of CA on mitochondrial morphology in the muscle using day-2-old young adults. We found that light-induced CA production from the gut bacteria does not affect muscular mitochondria. We conclude that (1) CA acts locally in intestinal cells, and (2) the protective effect of CA against age-associated fragmentation in muscular mitochondria that we observed previously (Han et al., 2017) is likely associated with the lifespan extending effect of CA, which may be secondary to the change in the intestine. We thank the reviewer for his/her suggestion, which help us distinguish the local effect of CA in the intestine and its systemic effect on other tissues.

5) This work relies on a non-standard gut microbe for *C. elegans* and requires that the animals be paralyzed to maintain colonization through measurement. This is addressed only in passing in the Discussion, but altogether still gives pause. Yes, *E. coli* is "this important Gram-positive model bacterium," but it isn't in fact a very important bacterium to this host. Limitations and open questions that should be made explicitly are: Whether host uptake of metabolites produced by a symbiont would differ from that of a food source, whether/how this approach could be extended to bacteria other than *E. coli* and how such interactions might play out in a community context and whether / how optogenetics could be used to study this question.

We appreciate the insightful suggestions from the reviewer to discuss the application of this new method in natural endosymbiont bacteria beyond *E. coli*. In the

Discussion, we note that we have recently demonstrated optogenetic control of gene expression in *Bacillus subtilis*, and suggest its possible application in studying Gram-positive bacterial-host interactions and longevity. Based on the reviewer’s suggestion, we have included additional discussions on 1) the difference in metabolite uptake from a symbiont and a food source, 2) the application of the *E. coli* optogenetic approach in host organisms where *E. coli* is a natural symbiont, 3) the development of new optogenetic approaches in different bacterial systems.

6) As a standard *C. elegans* feeding strain, OP50 is naturally lon deficient. Can the authors comment on the levels of CA production from their light-induced methods as compared to the regular OP50? If no such data is available, this should still be discussed.

Thank the reviewer for the suggestion. We haven’t directly compared to the level of CA from OP50 and the light-inducible bacterial strain. In the Han et al., 2017 paper, we reported that OP50 shows a 50% higher CA level than BW25113 and Δ*lon* shows a 4-fold higher CA level than BW25113. In the current work, we found that the CA levels from Δ*lon* and the light-inducible strain are similar (new Figure 4A). Based on this comparison, we conclude that the level of CA production from OP50 is lower than that from the light-inducible strain. We have included these information (subsection “Optogenetic control of CA production”).

7) The authors make a claim regarding the relative magnitude of lifespan effects of purified CA vs. CA produced using their optogenetic approach. However, it appears that claim is based on comparison of lifespan effects with previously published results. Such comparisons are risky because differences in lab, feeding conditions, and strain. It would also have been preferable to have a CA "fed" control and it would further have been interesting to carry out a study of optogenetic induction in the presence of exogenous CA. In the absence of a CA fed control, the authors should be careful with their claims regarding effect size.

We appreciate the reviewer’s comment. In our experiments (Figure 4), we have chosen the Δ*lon* mutant as a positive control and a comparative reference for the effect of CA produced using the optogenetic approach, since the lifespan-extending effect of Δ*lon* is stronger than CA supplementation and CA supplementation does not enhance the lifespan extension by the loss of *lon* (as shown in the Han et al., 2017 paper). In Figure 4 and Supplementary file 1C, all the lifespan assays were conducted in parallel for direct comparison. We found that the lifespan extension caused by the light-inducible strain is comparable to that caused by Δ*lon*. Based on light-inducible = Δ*lon* (in this work) and Δ*lon* > CA (Han et al., 2017), we concluded that light-inducible > CA. We agree with the reviewer that we should be more careful with our claims, since we did not directly include CA supplementation as a control. We have revised text in the Introduction and Results to state this claim more precisely.

8) There were some concerns over the exclusive use of GFP with a mito targeting sequence as the reporter because this only labels mitochondria if they are functionally importing the GFP. In addition there seems to be a lot of noise from intestinal gut granules and aggregates in Figure 3B. These data could be improved if an additional label of mitochondria was used – either a membrane marker or dye, in order to ensure the changes seen are reflecting changes to mitochondrial network morphology and not GFP aggregation, or non-mitochondrial localized GFP signal. The authors should be careful to make these limitations clear.

We appreciate the reviewer’s suggestion, and we have conducted this experiment using another RFP reporter that localized to the mitochondrial membrane, in which the Nterminal 55 amino acids sequence of *C. elegans tomm-20* is fused with RFP. Using this new reporter, we have validated the results using mito-GFP (new Figure 3C, E and Figure 3—figure supplement 1).

9) Lifespan of control and treated animals in general appear shorter than would be expected. E.g. in figure 4, the lifespan of worms fed the *E. coli* background strain JW1935-1 is considerably short. Can the authors comment on any previous studies with *C. elegans* lifespan on this strain and if this is expected? Are there other optogenetic background strains available?

Thank the reviewer for pointing this out. We are sorry that we did not explain this experimental design more clearly. The lifespan assay was conducted at 26 degrees using the *sqt-3(e2117)* temperature sensitive mutant that reproduces normal but is embryonically lethal at 26 degrees. In this way, we can perform the experiment at the restrictive temperature to avoid animal transfer that causes white light exposure, while not using floxuridine/FUDR that interrupts normal reproduction and alters bacterial metabolism. We have used this method previously in the Han et al., 2017 paper, and the mean lifespan is 8.7 ± 0.1 days. Thus, the lifespan, ~6.5 days, shown in Figure 4 is slightly shorter. We think this is not due to the *E. coli* strain background given that the lifespan of Δ*lon* is also shorter, ~ 9 days (this work) vs. ~11 days (in Han et al., 2017). The possible explanation is that light exposure may shorten the lifespan of *C. elegans* (PMID: 29500338). In the revised manuscript, we have explained the experimental design more clearly (subsection “Lifespan experiments”) and included the temperature information in figure panels (new Figure 4B-D).